

# The diel vertical migration of microbes within snowpacks driven by solar radiation and nutrients

Masato Ono[1], Nozomu Takeuchi[2]

[1]Center for Ecological Research, Kyoto University, Shiga 520-2113, Japan

[2]Department of Earth Science, Graduate School of Science, Chiba University, Chiba, 263‑8522, Japan

*Correspondence to*: Masato Ono (ono.masato.2v@kyoto-u.ac.jp)

**Abstract.** Seasonal snowpacks are inhabited by various microbes despite their ephemeral and cold environments. Physicochemical conditions in snowpacks drastically change with day-night cycles; however, their effect on the microbial community is largely unknown. This study describes the diel vertical migration (DVM) of microbes within a snowpack in an

alpine forest in northern Japan. Microscopy revealed the presence of snow algae, microinvertebrates, and fungi in the snowpack. Periodic sampling across snow depths revealed that the vertical distribution of microbes changed over time. Motile cells of snow algae and microinvertebrates were distributed near the surface at night but migrated to a depth of 10–20 cm during the day. Other microbes, including algal spores and fungi, remained on the surface layer throughout the day. The vertical migration of microbes was synchronized with the intensity of solar radiation, suggesting that the microbes moved downward to avoid

intense solar radiation on the snow surface. Soluble nutrients ($PO_4^{3-}$, $NH_4^+$, and $K^+$) were always the highest at the snow surface, suggesting that surface snow provides a favorable environment for microbial growth. These results indicate that motile microbes migrate diurnally within the snowpack to remain under the best conditions for their growth in terms of the intensity of solar radiation and nutrients. Snow core sampling from the surface to bottom revealed that microbes were mostly present above a depth of 30 cm in the snowpack. Therefore, the upper snow layer above this depth, referred to as the Microbial Active

Snow Surface (MASS) layer, is important for the life cycles of microbes and biogeochemical cycles within snowpacks.

## 1 Introduction

Seasonal snowpacks cover up to 30% of the global land surface and are inhabited by various cold-tolerant organisms. They are known as snow-ice organisms and have been reported mainly from polar to alpine regions (Fukushima, 1963; Hoham and

Duval, 2001; Hoham and Remias, 2020). Photosynthetic microbes such as snow algae grow on snow surfaces and supply organic matter to animals, fungi, and bacteria living in snowpacks. These animals include tardigrades, rotifers (Hanzelová et al., 2018; Yakimovich et al., 2020; Ono et al., 2021, 2022), springtails (Hao et al., 2020), and winter stoneflies (Negoro, 2009). They feed on algae and redistribute organic matter as they migrate through the snowpack. Heterotrophic bacteria and fungi are also active in snowpacks, although information on their life cycles is limited (Irwin et al., 2021; Matsuzaki et al., 2021;



Nakanishi et al., 2023). Because these organisms are tropically associated with each other (Brown et al., 2015; Ono et al., 2021), snowpacks can be acknowledged as unique ecosystems (Domine, 2019).

Snow cover extent (SCE) has significantly decreased recently as a result of an increase in global mean air temperature (McCabe and Wolock, 2010), indicating the shrinkage of habitats for snow and ice organisms. A climate model combing an observation-based ensemble showed that the SCE of the northern hemisphere decreased by $-1.9 \times 10^6$ km °C$^{-1}$ over the 1981–

2010 period (Mudryk et al., 2017). In Japan, numerical experiments have estimated that snow depth will decrease by more than 60% by the 2070s (Hara et al., 2008). The climate model projected that the timing of snowmelt in the mountainous areas of central Japan would begin half a month earlier than the present climate when the global air temperature is 2 K warmer than that in the pre-industrial period and that the snowpack would disappear two months earlier when it is 4 K warmer (Kawase et al., 2020). Therefore, it is crucial to predict future changes in snowpack ecosystems under climate warming conditions.

The activities of snow-ice organisms are largely affected by the physicochemical conditions of the snowpack. Previous studies have suggested that the presence of meltwater and intensity of solar radiation determine the distribution of snow algae in snowpacks. As snow algae require meltwater for germination (Hoham, 1975), they are often concentrated in layers with abundant meltwater (high liquid water content) within the snowpack (Hoham, 1975; Grinde, 1983). The intensity of solar radiation affects not only the snowpack melt rate, but also the photosynthesis and life cycle stages of snow algae. For example,

the mating of *Chloromonas* mainly occurred under irradiation of 95 µmol PAR m$^{-2}$ s$^{-1}$ which is equivalent to 19 W m$^{-2}$ (assuming incandescent as a light source as referenced from Thimijan and Heins (1983), the same conversion applies to all the following)  (Hoham et al., 1998). The cyst of green snow algae (*Chloromonas hindakii*) adapted to higher irradiation up to 600 µmol PAR m$^{-2}$ s$^{-1}$ (equivalent to 120 W m$^{-2}$), while photoinhibition occurred when the intensity of light was more vigorous (Procházková et al., 2019a).

Nutrients are also relevant to the distribution of snow algae (Tranter et al., 1987) as they are essential for their growth, in addition to meltwater and solar radiation. Previous studies showed that the growth of snow algae is promoted by high concentrations of NH$_4^+$ (Novis, 2002), and abundant snow algae were observed in the snowpack with high NO$_3^-$ concentration (Hanzelová et al., 2018). In contrast, another study showed that the concentrations of nutrients in snowpacks with snow algae were lower due to their consumption (Jones, 1991). The heterogeneous distribution of nutrients in snowpacks has been

explained by precipitation, snowfall, and the deposition of mineral particles and plant litter on the snow surface (Hoham, 1976; Jones, 1991).

The source of snow algal cells and the behavior of algal consumers may also affect their distribution within the snowpack. Snow algae are probably derived from the atmosphere and/or soil beneath the snowpack. If snow algal cells are transported from the atmosphere, they accumulate on the snow surface. A snow alga, *Sanguina nivaloides* (previously known as

*Chlamydomonas nivalis*) (Procházková et al., 2019b), was suggested to be transported on the snow surface by wind (Marshall and Chalmers, 1997). Flagellated motile algal cells can migrate from the soil under snowpacks to the surface (Hoham and Duval, 2001). Tardigrades and rotifers have often been observed feeding on snow algal cells in snowpacks (Ono et al., 2021)



and may affect the distribution of snow algae, although there have been few reports on the association between snow algae and consumers.

Information on the vertical distribution of snow-ice microbes within snowpacks is limited because most studies have focused on the snow surface (0–2 cm in depth) as their habitat. Snow algae, which require light for photosynthesis, may prefer the surface layer of snowpacks. However, snow algae are sometimes found in deeper layers of snowpacks (Schuler and Mikucki, 2023). For instance, red snow algae are distributed in layers down to a depth of 10 cm in snowpacks in North America (Thomas, 1972). Green snow algae were distributed along the ice layers, which formed when meltwater percolated from the refrozen

snow surface (e.g., 8–12 cm in depth, Hoham, 1975).

Photoplankton and zooplankton in aquatic environments such as lakes and oceans exhibit diel vertical migration (DVM). Their DVM was driven by the upwelling of water, intensity of solar radiation, and water temperature. For example, higher water temperatures cause earlier downward migration of the freshwater alga *Gonyostomum semen* (Rohrlack, 2020). Similarly, the zooplankton *Daphnia* showed upwelling swimming when the intensity of light continuously decreased (Ringelberg, 1964).

The DVM of phytoplankton was also affected by consumer behavior (Reichwaldt and Stibor, 2005). Although these aquatic environments are physically different from those of snowpacks consisting of solid ice with limited meltwater, microbes living in snowpacks may have DVM in response to diel changes in meltwater and solar radiation. Green snow from algal blooms has been observed to become visibly pale in color as solar radiation increases (Kawecka, 1986). In addition, an irradiation experiment revealed that snow algae migrate to deeper layers within snowpacks in response to changes in light intensity

(Grinde, 1983). However, quantitative studies on the DVM of snow-ice microbes are yet to be conducted.

In this study, the 24 h periodic sampling across snow depths was carried out to describe the DVM of microbes within a snowpack in a forest area in northern Japan, where various algae and microinvertebrates have been reported (Muramoto et al., 2008, 2010; Tanabe et al., 2011; Matsuzaki et al., 2014, 2015; Ono et al., 2021, 2022; Nakanishi et al., 2023; Suzuki and Takeuchi, 2023). Changes in the vertical distribution of each microbe are discussed, along with diel changes in the physical

and chemical solutes in the snowpack.

## 2 Material and Methods

### 2.1 Study site and sample collection

Sample collection was conducted at Yumiharidaira Park (38° 30′ N, 140° 00′ E; 770 m above sea level (a.s.l.)) on Mt. Gassan, Yamagata prefecture, Japan (Fig. 1a). Mount Gassan has an asymmetric shape, represented by a horseshoe-shaped caldera

(Nakazato et al., 1996). During winter, strong westerlies from Siberia, with moisture from the Sea of Japan, blow directly to Mt. Gassan, resulting in heavy snowfall on this mountain range (Kariya, 2002, 2005). Snowpack monitoring have been conducted in this area ("Shizu" located at 710 m a.s.l.) by the National Research Institute for Earth Science and Disaster Resilience (NIED) since 1995. The average temperature during the winter season (December 1st to March 31st) from 1995 to





2010 at this site was -3.8 °C, with an average maximum snow depth of 3.6 m, which is equivalent to 1358 mm in water

(Yamaguchi et al., 2011). The snowpack began to melt in March and continued until early summer.

The timber line in this area is located between 1400 and 1500 m a.s.l. Below this elevation, deciduous and coniferous trees, including beech trees, are dominant. Above the timberline, the vegetation consists mainly of dwarf forests (Kariya, 2005).

Meteorological data were recorded for the forested area during the study period (Fig. 1b). The air temperature was recorded every 10 min with a temperature/humidity data logger (TR-72nw, T&D Corporation, Japan), maintained at a height of 50 cm

from the snow surface by installing it in a 3 m probe. The intensity of solar radiation was recorded every 10 s using a pyranometer (ML-020VM, EKO, Japan) with a data logger (LR5091, HIOKI, Japan) at a height of 5 cm from the snow surface by setting it on a small pedestal. The recorded air temperatures and solar radiation were converted into hourly averages.

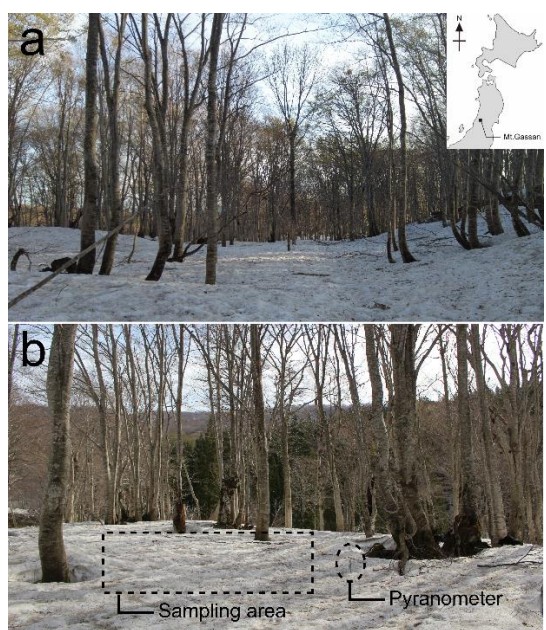

**Figure 1: Study site and sampling area on May 6th, 2021: (a) study site, (b) sampling area, and the install point of a pyranometer.**

To describe the vertical distribution of microbes and chemical solutes within the snowpack, snow samples from a forest area dominated by beech trees were collected every three hours from 2:00 am on May 6th to 2:00 am on May 7th, 2021 (Japan Standard Time). Beech trees did not open their leaves. Patches of green snow were observed on the snowpack surface during the study period, as previously reported (Fig. 2a) (Ono et al., 2021, 2022). Snow depths before and after periodic sampling were 105 cm (5:00 am on May 6th) and 98 cm (5:00 am on May 7th), respectively. The weather was clear without clouds

throughout the study period. Snow samples from an area of $5 \times 5$ cm were continuously collected in five layers across snow depths (0–3 cm in depth: Layer I, 3–8 cm: Layer II, 8–13 cm: Layer III, 13–18 cm: Layer IV, and 18–23 cm: Layer V, Fig. 2b). Sampling was performed on three different surfaces at each time point. All snow samples (in total 135 samples) were collected using a small spatula and preserved in Whirl–Pak bags (B01065WA, Nasco, USA). In this study, the period from 0:00 on May 6th to 0:00 on May 8th was represented as 0:00 to 48:00.



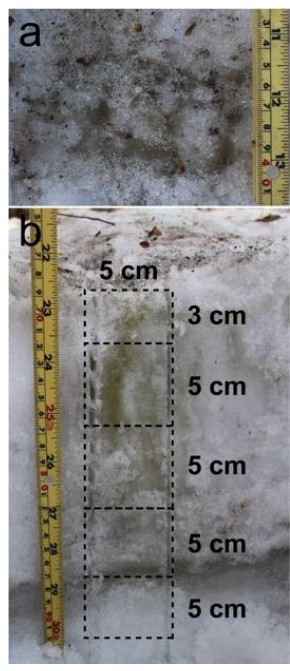


**Figure 2: Conditions of the snowpack on May 6th, 2021: (a) snow surface, (b) cross-section of the snow pit. Periodic sampling was conducted across the snow depth divided into five layers: layer I (5 × 5 × 3 cm in length × width × height), layer II (5 × 5 × 5 cm), layer III (5 × 5 × 5 cm), layer IV (5 × 5 × 5 cm), and layer V (5 × 5 × 5 cm).**

A snow core was collected from the surface to the bottom of the snowpack using an ice auger to determine the vertical

distribution of microbes throughout the snow layer (Fig. S1 in Supplementary Information). Snow core samples were collected

at 5:00 a.m., immediately after sunrise on May 7th. The core, with a total length of 113 cm, was cut horizontally every 10 cm

using a snowsaw and preserved in Whirl–Pak bags (in total 12 samples).

Time-lapse images of the snow surface in the study area were captured using a 1.3 M pixel interval camera (TLC200 Pro,

Brinno, Taiwan). An interval camera, at a height of 1 m from the snow surface, was installed on a 1.5 m stainless pole. The

images were captured every 10 min from 2:00 a.m. on May 6th to 8:00 a.m. on May 7th, except for the time without light,

which was mostly from 8:00 p.m. to 4:00 a.m. Captured images were saved as JPEG files of 1280 × 720 pixels and then output

as an AVI file.

## 2.2 Sample processing

The collected samples were frozen and transported to the laboratory at Chiba University, Japan. The samples were stored in a

freezer (-20 ℃) until further processing. The samples were slowly melted in a refrigerator (5 ℃) before analyses. After melting,

the samples were separated into 10 mL for snow algae and fungi, 5 mL for soluble ion analysis, and the remainder for

microinvertebrates. The sub-samples for snow algae and fungi were stored in 10 mL plastic tubes with 3% formaldehyde. The

samples for chemical analysis were put in 6 mL plastic tubes after filtering the meltwater through a 0.45 μm ion-free disposable

filter (13AI Chromatodisk, GL-science, Japan) to remove microbes and particulate dust.



## 2.3 Measurements of chlorophyll a and microscopic observations

Chlorophyll a concentration in the snow samples was measured and used as a proxy for algal biomass. Chlorophyll a is a photosynthetic pigment found in phototrophs that is commonly used as an indicator of the total biomass of phototrophs (Holm-Hansen et al., 1965). Chlorophyll a concentration ($\mu$g L$^{-1}$) was calculated by measuring fluorescent intensity of extracted pigment. A further method has been described by Ono et al. (2021).

Microscopic observations were conducted to describe the morphology of the snow algae and fungi. Twenty microliters of the sample were transferred from the subsample onto a glass slide. Images of algae and fungi were captured using a fluorescence microscope (BX51, Olympus, Japan) equipped with a digital camera (DP21, Olympus, Japan). The geometric size of each algal and fungal cell (long and short axes of algal cells and the length and thickness of protrusions of fungal cells) was measured using image processing software (Image J 1.38X, National Institutes of Health, USA). Algal species were not identified in this study, because morphological taxa are not always represented as phylogenetic species of snow algae.

The cell numbers of snow algae and fungi in the snow samples were counted to calculate cell concentrations. 5–200 $\mu$L from the subsamples were injected into a filter holder equipped with a 0.45 $\mu$m PTFE membrane filter (JHWP01300, Merck Millipore, Germany), then filtered using a pump (Linicon LV-125, Nitto Kohki, Japan). The volume injected into each sample was adjusted based on the chlorophyll a concentration. The filter was then placed on a slide and covered with a cover glass and Milli-Q water. Cell counts for all snow algae and fungi observed on the filter were performed three times for each sample using a fluorescence microscope. The average cell number and the sample volume used for filtration were used to calculate the cell concentration per water equivalent of the snow sample (cells L$^{-1}$). Their vertical distributions are represented as proportions of cell concentrations in each layer of the snow pit or snow core.

The number of tardigrades and rotifers in the snow samples was counted to calculate the population density of each snow layer. The melted snow sample remaining after processing was moved to a petri dish to count tardigrades and rotifers. Tardigrades and rotifers were counted under a stereomicroscope (MZ125; Leica Microsystems, Germany). The population density of the microinvertebrates (ind L$^{-1}$) was calculated using the number of observed microinvertebrates and the water equivalent of the snow sample. Their vertical distributions were represented as the proportion of individual concentrations in each layer to the total concentration in the snow pit or snow core. A portion of the tardigrades and rotifers was mounted on slides in a small drop of Hoyer's medium and observed under a phase-contrast microscope (BX51, Olympus, Japan) to identify the species. Species identification keys were used for tardigrades (Ono et al., 2022) and rotifers (Wallace et al., 2015).

## 2.4 Analysis of major chemical solutes

The concentrations of major chemical solutes, including $Na^+$, $Cl^-$, $K^+$, $Ca^{2+}$, $Mg^{2+}$, $SO_4^{2-}$, $NH_4^+$, $NO_2^-$, $NO_3^-$, and $PO_4^{3-}$, were analyzed with an ion-chromatography system (for anion: AQUION, Thermo Fisher Scientific, USA, for cation: ICS-1100, Thermo Fisher Scientific, USA). Concentrations were given in equivalent units per melt water of snow ($\mu$eq Kg$^{-1}$). In this study, $NO_2^-$ was below the detection level in all samples.



## 2.5 Statistical analysis

A chi-square test was performed to test for differences in the vertical distributions of microbes and chemical solutes across the study period. Pearson's correlation coefficients were obtained between the microbes and meteorological conditions (air temperature and solar radiation) or each chemical solute. Additionally, cluster analysis was performed to group the chemical components using the Euclidean distance and Ward methods. All statistical analyses were performed using the R software (R Core Team, 2022).

## 3 Results

### 3.1 Meteorological conditions during the study period

The air temperature recorded at the study site ranged from 2.5 to 21.7 °C (Fig. 3a). The lowest air temperature (2.5°C) was recorded at 2:40 in nighttime and then slightly fluctuated (2.5–4.3 °C) until sunrise. It increased after 6:00 and reached the highest (21.7 °C) at 15:10, and then gradually decreased down to 2.6 °C by 25:10. Solar radiation ranged from 0 to 755 W m$^{-2}$ (Fig. 3b). The intensity of solar radiation started to increase at 6:00, reached its peak (755 W m$^{-2}$) at 14:00, and then decreased continuously until 20:00 (0.6 W m$^{-2}$). In this study, daytime was defined as the period from 9:00 to 16:00 when the intensity exceeded 150 W m$^{-2}$, and nighttime was defined as the remaining period of the day.

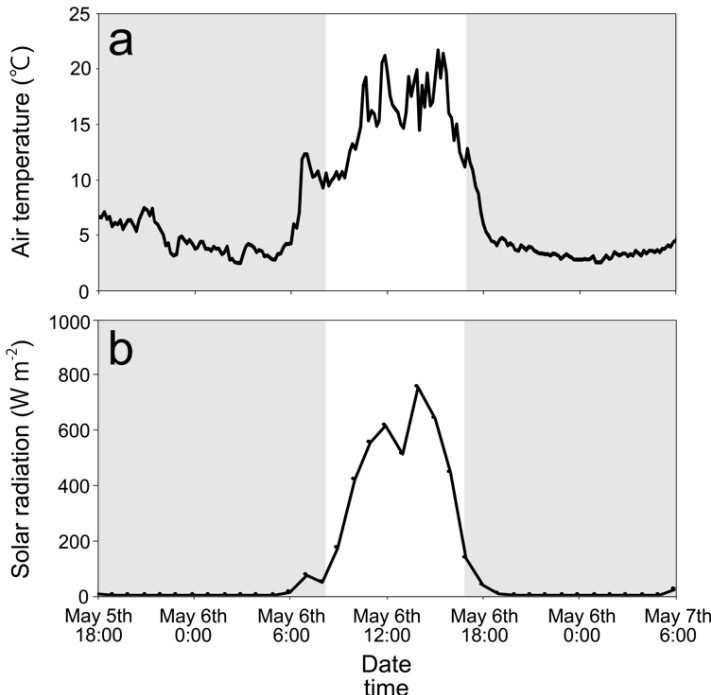

**Figure 3: Meteorological conditions recorded during the study period. (a) air temperature, (b) solar radiation. The period of nighttime is shown as gray zones.**





## 3.2 Snow-ice microbes in snowpack

Microscopic observations revealed that various microbes (snow algae, microinvertebrates, and fungi) inhabit the snowpack (Fig. 4). Three morphological types of snow algae were dominant in the snowpack (snow algae types A–C) (Fig. 4a–d). Type A cells were spherical with green chloroplasts. There were two different sizes in this type of algae; small ones ranged from 4.6 to 9.9 μm (mean ± SD: 8.7 ± 1.5 μm, n = 245), and large ones ranged from 10.0 to 22.6 μm (14.8 ± 2.4 μm, n = 372) in length, respectively (Fig. 4a, b). Type B had oval-shaped cells with green chloroplasts (Fig. 4c). The length of this algae ranged from

13.7 to 23.1 μm (18.6 ± 1.4 μm, n = 313) and the width from 7.5 to 13.1 μm (10.0 ± 1.0 μm, n = 313). Type C has oval-shaped cells with ribbed walls, green chloroplasts, and orange secondary carotenoids (Fig. 4d). The length of this algae ranged from 25.4 to 54.2 μm (34.3 ± 4.7 μm, n = 313) and the width from 12.8 to 29.9 μm (20.8 ± 2.2 μm, n = 313).

  Two genera (Tardigrada and Rotifera) of microinvertebrates and snow fungi were also frequently observed in the snowpack (Fig. 4e–g). Two species of tardigrades, *Hypsibius* sp. and *Hypsibius nivalis* (Ono et al., 2022), featured their asymmetrical

claws dominated (Fig. 4e). Identifying the rotifer species was difficult because of the absence of live species; however, Philodinidae dominated (Fig. 4f). A significant number of fungal cells were observed in these samples. The fungal cell had 3– 5 protuberances and a substrate in their center (Fig. 4g), which is likely to be *Chionaster nivalis* (*Chi. nivalis*) as reported in Matsuzaki et al. (2021).

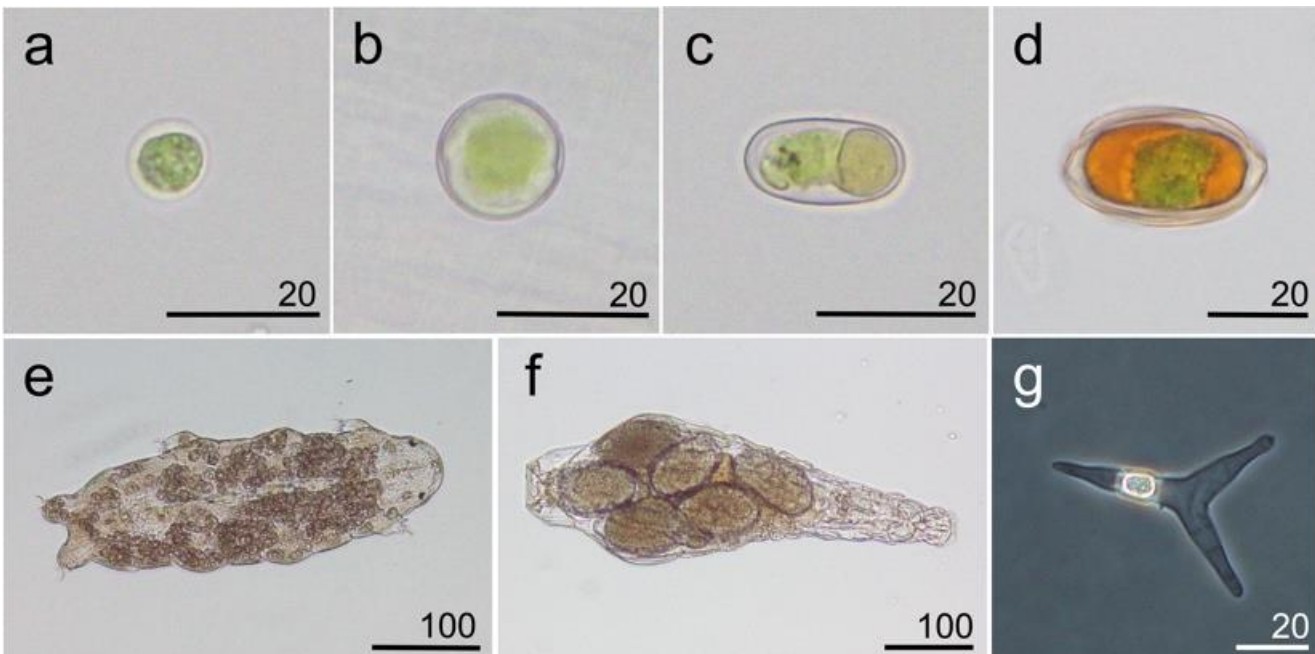

**Figure 4: Snow-ice microbes inhabit the snow. (a), (b) Snow algae Type A (LM), (c) Snow algae Type B (LM), (d) Snow algae Type C (LM, dormant state), (e) *Hypsibius* spp. (LM), (f) Philodinidae gen. sp. (LM), (g) *Chionaster nivalis* (PCM). All scale bars are in micrometers.**





### 3.3 The vertical distribution of snow-ice microbes in the snowpack

Periodic sampling across various snow depths revealed that certain microbes exhibited alterations in their vertical distribution

within the snowpack, whereas others maintained a consistent distribution (Fig. 5, S2, S3, Table S1). The vertical distribution

of chlorophyll a, cell concentrations of *Chloromonas* sp. Types A and B and the population density of tardigrades and rotifers

changed from nighttime to daytime. In contrast, the distribution of other microbes (*Chloromonas* sp. Type C and *Chionaster*

*nivalis*) remained unchanged throughout the study period.

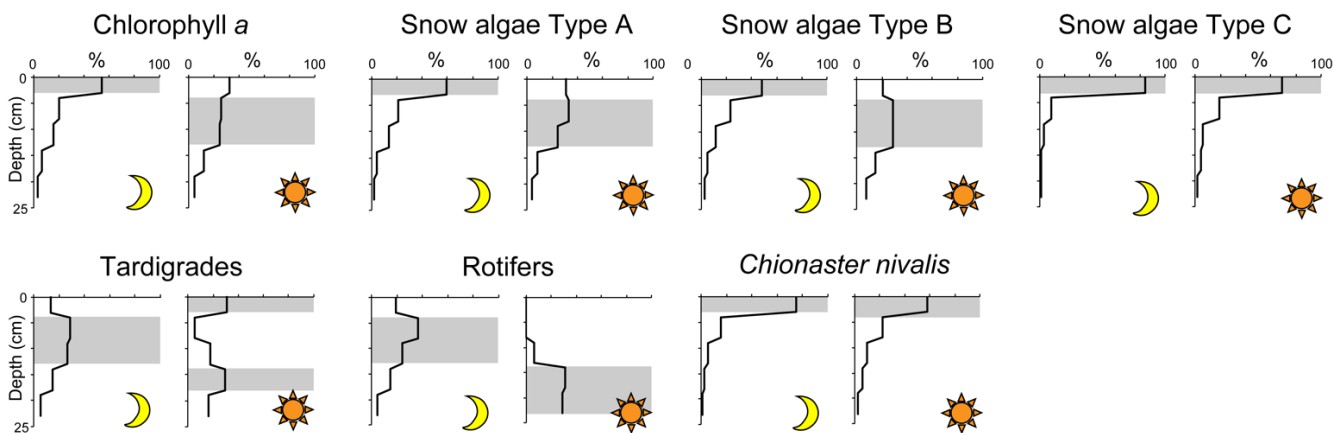

**Figure 5: The vertical distribution of microbes in nighttime and daytime. %: Percentage of chlorophyll a concentration, cell concentration, and population density of each layer in the total of five layers. The layers where the microbes were concentrated are shown as gray zones.**

### 3.3.1 Snow algae

*Chlorophyll a*

Chlorophyll a in the snowpack showed a heterogeneous distribution across layers, and the layer with the highest concentration

changed over time (Figs. 5, S2, Table S1). The highest concentration occurred in the uppermost surface layer (layer I) between

2:00 and 8:00. It was then broadly distributed between layers I and III from 11:00 to 14:00. The highest concentration was

observed in layers I and II during the night from 17:00 to 26:00. The mean chlorophyll a concentration in the layer I was 4.9

$\pm$ 2.6 $\times$ $10^2$ µg L$^{-1}$. The difference of the vertical distributions between nighttime and daytime was statistically significant ($\chi^2$

(4) = 9.92, p = 0.04 < 0.05) (Table S2).

*Algal community structure and cell concentrations of algal taxa*

The algal community structure in the snowpack differed between the nighttime and daytime (Fig. 6). The proportion of each

algal type in the total algal biomass in layer I showed that Type B was dominant during nighttime, whereas Type C was

dominant during daytime. The algal community in layer I was dominated by Type B (48.4−60.8%), followed by Types A

(16.5−33.4%) and C (10.1−23.1%) during the nighttime. The dominant algae changed to Type C (49.5%), followed by Type



B (29.3%) and Type A (21.2%) in the morning at 8:00. During the daytime, Type C was the most dominant (76.1−84.4%) while Types A and B were less dominant (7.2−15.7% and 8.2−8.4%, respectively). The algal communities in the lower layers (layers II–V) were dominated by Types A and B throughout the study period.

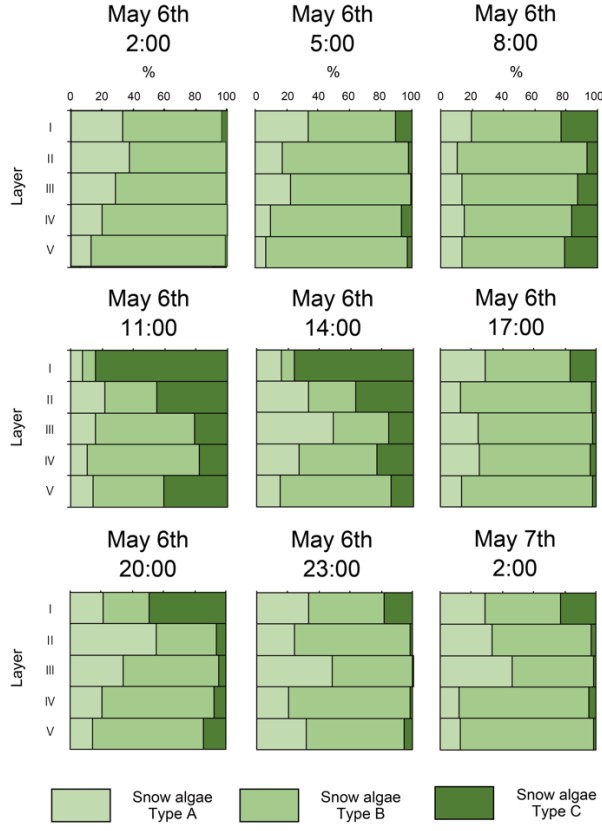

**Figure 6: The community structure of snow algae in each layer every three hours.**

*Cell concentrations of algal taxa*

Temporal changes in the vertical distribution of cell concentrations varied among snow algal taxa. The vertical distributions of type A and B algae were heterogeneous across layers, and the layer with the maximum population changed over time. In contrast, Type C showed the maximum concentration in Layer I throughout the study period (Figs. 5, S2, Table S1). The vertical distributions of types A and B were similar to those of chlorophyll a, showing the maximum concentration in layer I from 2:00 to 8:00. They were distributed between layers I and III from 11:00 to 14:00, but were distributed either in layers I or II from 17:00 to 26:00. The mean cell concentrations of Type A, Type B, and Type C at layer I were $1.9 \pm 1.1 \times 10^7$ cells L$^{-1}$, $1.8 \pm 1.1 \times 10^7$ cells L$^{-1}$, $1.6 \pm 1.0 \times 10^6$ cells L$^{-1}$, respectively. The differences in the vertical distributions of Types A and B between the nighttime and daytime were statistically significant (Table S2).



### 3.3.2 Microinvertebrates

Temporal changes in the vertical distributions of tardigrades and rotifers showed similar patterns. Their distributions were heterogeneous across the layers, and the depth at which the maximum population occurred changed over time (Figs. 5 and S2; Table S1). The maximum populations of tardigrades and rotifers were found in layers I, II, and III, and layers I, II, III, and IV

from 2:00 to 8:00, respectively. Tardigrades were concentrated at two different depths (layers I and IV to V) from 11:00 to 14:00, and were then observed in either layer I or layer II from 17:00 to 26:00. Rotifers were concentrated in layers IV to V from 11:00 to 14:00, and were then observed in layers II, III, or IV from 17:00 to 26:00. The mean population density in layer I was $3.5 \pm 4.4 \times 10^3$ ind L$^{-1}$and $5.1 \pm 8.2 \times 10^2$ ind L$^{-1}$, respectively. Significant differences in the vertical distribution of microinvertebrates were observed between the nighttime and daytime (Table S2).

### 3.3.3 Snow fungi

*Chi. nivalis* in the snowpack showed the maximum concentration at layer I throughout the study period (Figs. 5, S2, Table S1). The cell concentration was the highest in layer I and decreased exponentially with depth. The average cell concentration at layer I was $1.6 \pm 1.0 \times 10^6$ cells L$^{-1}$. There were no significant differences in the vertical distribution of *Chi. nivalis* between daytime and nighttime ($\chi^2$ (4) = 7.10, p = 0.13 > 0.05) (Table S2).

### 3.4 Snow-ice microbes in a snow core

Microscopic analysis of the snow core revealed that all the microbes were distributed above a depth of 30 cm (Fig. S3). The population of each microbe above a depth of 30 cm accounted for over 95% of the total population in the snow core (chlorophyll a: 96.3%; *Chloromonas* Type A 98.5%; Type B: 99.4%; Type C: 95.6%; tardigrades: 99.7%; rotifers: 100%; *Chi. nivalis*: 97.3%).

### 3.5 The vertical distribution of chemical solutes in the snow pit

The vertical profiles of the major chemical solutes in the snow pit showed that the dominant solutes differed between the upper (I, II, and III) and lower layers (IV and V) (Figs. 7, S4). The layer I was dominated by NH$_4^+$ (23.5%), PO$_4^{3-}$ (20.8%), K$^+$ (19.8%), Cl$^-$ (16.0%), and Na$^+$ (11.6%). The layers II and III were dominated by Cl$^-$ (28.1−29.8%), Na$^+$ (23.1−25.2%), NH$_4^+$ (11.2−12.5%), and PO$_4^{3-}$ (8.7−10.6%). The lowest layers (IV and V) were dominated by Cl$^-$ (33.8−35.8%), Na$^+$ (28.8−29.0%),

and NH$_4^+$ (8.9−10.1%) (Fig. S4).

The chemical solutes in the snow samples were divided into two groups based on their vertical distribution (Fig. S5). There were significant positive correlations among three solutes (NH$_4^+$, PO$_4^{3-}$, and K$^+$) and among the other solutes (Na$^+$, Cl$^-$, SO$_4^{2-}$, NO$_3^-$, Mg$^{2+}$, and Ca$^{2+}$) (Fig. S5). The vertical profiles of the solute concentrations were distinct between the two groups (Figs. 6 and S6). The concentrations of NH$_4^+$, PO$_4^{3-}$, and K$^+$ were the highest in the surface layer, while those of the others were the





highest in the deeper layers of snow throughout the study period. There was no significant difference in the vertical distribution of these solutes between the nighttime and daytime (Table S2).

Figure 7: The vertical distribution of chemical solutes in nighttime and daytime.

**3.6 Snow surface observation with a time-lapse camera**

Time-lapse photographs of the snow surface revealed that the color of snow surface was changed drastically over time (Fig. 8). A green snow patch (15 cm × 5 cm) was observed at 2:10 nighttime (square with a white dashed line in Fig. 8). The green color gradually disappeared by 9:00 in morning and remained until 17:20. The green color appeared again on the snow surface at 18:00 and remained until 30:00.





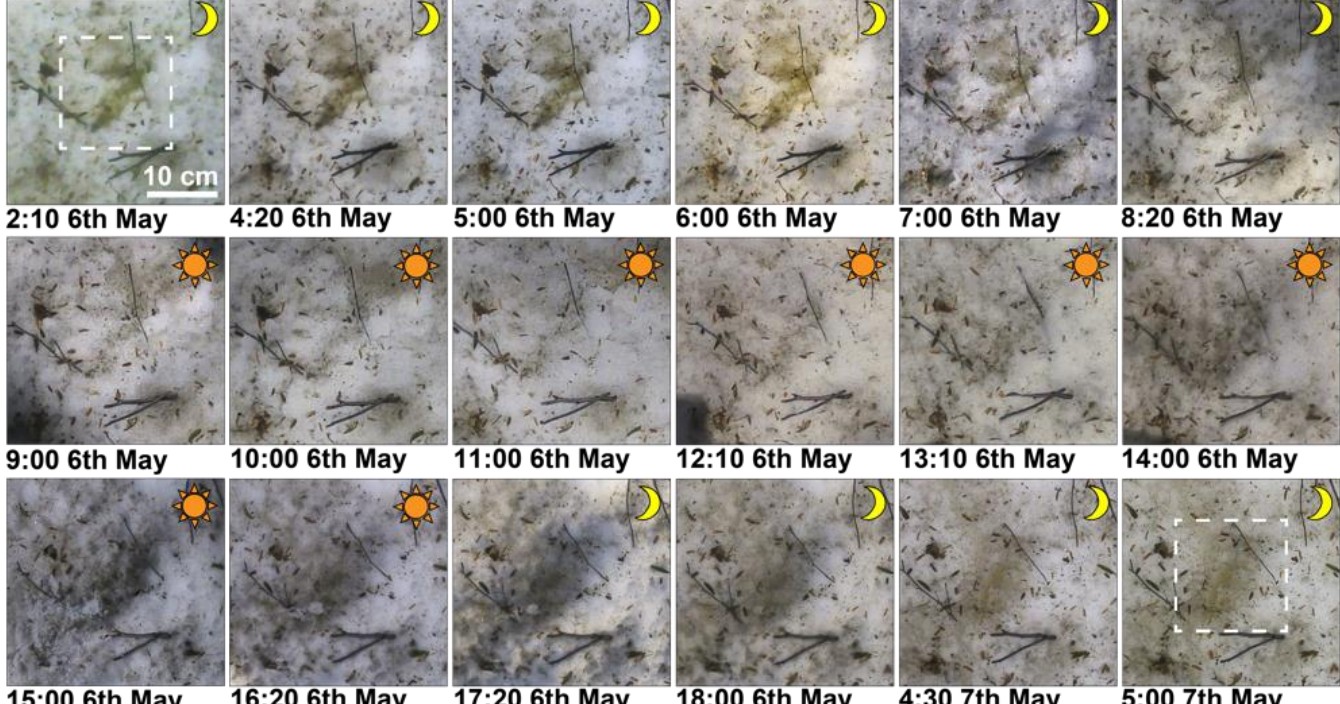

**Figure 8: Photographs of snow surface captured by a time-lapse interval camera during the study period. A green snow patch can be seen within the white square at 5:00 on May 6th and 6:00 on May 7th. The scale bar is shown in an image taken at 5:00 on May 6th.**

## 4 Discussion

### 4.1 Vertical distribution of motile and immotile microbes

The 24 h periodic samplings of snow-ice microbes revealed that microbes could be divided into two groups: those with and without diel changes in the vertical distribution. The vertical distribution of microbes with diel changes was significantly different between the nighttime and daytime. This group includes Types A and B snow algae, tardigrades, and rotifers. Microbes without a diel change maintained their maximum population in the surface layer of the snowpack throughout the study period. These included Type C snow algae and *Chi. nivalis*. This difference in diel activity can be explained by the motility of the microbes. The former group consists of motile microbes, while the latter group consists of immotile cells.

The maximum population of microbes at the surface layer without diel change suggests that they passively accumulated at the surface due to atmospheric deposition and/or concentrated at the surface as the snowpack melted. The microbes without diel changes included Type C algae and *Chi. nivalis*, which are a dormant algal cell and a fungus, respectively, and both of them are immotile. The vertical distribution of immotile microbes is likely determined by their source, propagation, and passive movement in the snowpack. Type C algae may also be derived from the soil surface below the snowpack as they change from





motile vegetative cells, which are possibly Type B algae (Muramoto et al., 2008; Matsuzaki et al., 2019; Procházková et al., 2019a).

The most abundant populations of microbes in the surface layer of the snowpack throughout the study period indicate that percolating meltwater did not wash these microbes down to the lower layers in the snowpack. Meltwater is produced by the
melting of the snow surface, mainly during the daytime, and percolates downward in the snowpack. If meltwater affects the distribution of snow-ice microbes, immotile microbes should be widely distributed within the snowpack. Vertical distributions showed that the two species of microbes were concentrated in Layer I throughout the study period (Figs. 5 and S2, Table S1). This is consistent with previous studies showing that microbes within a snowpack are rarely washed by percolating meltwater down to the lower layers, but remain on the snow surface (Marchant, 1982; Grinde, 1983; Hoham et al., 1993). Therefore,
immotile microbes in the snowpack are unlikely to be moved passively by meltwater but are concentrated in the surface layer of the snowpack as snowmelt proceeds.

### 4.2 Factors determining vertical distribution of motile microbes

Multiple factors determine the vertical distributions of motile microbes. Solar radiation and nutrient availability in the snow layers are likely to be effective factors for algae, whereas solar radiation and the distribution of food resources are likely to be
important factors for tardigrades and rotifers.

### 4.2.1 Factors determining vertical distribution of motile snow algae

The timing of the movement of motile snow algae coincided with changes in the intensity of solar radiation, suggesting that snow algae moved downward away from the intense solar radiation at the snow surface. The position of the maximum population of motile snow algae changed from layer I to layers II and III at 8:00 when the solar radiation reached 170 W m$^{-2}$.
After the maximum intensity of solar radiation at 14:00 (755 W m$^{-2}$), the position of the maximum population changed again to the upper layer (layer I), suggesting that the algae moved upward over time. The photographs captured by the time-lapse interval camera consistently showed a diel change in color on the snow surface. The green color of the snow surface began to disappear at 8:20 (Fig. 8). The population of algal cells at the surface layer (layer I) showed a significant negative correlation with both the intensity of solar radiation and air temperature (Type A and solar radiation: r = -0.44, p = 0.02 < 0.05; Type A
and temperature: r = -0.49, p = 0.01 < 0.05; Type B and solar radiation: r = -0.64, p = 0.00 < 0.05; Type B and temperature: r = -0.48, p = 0.01 < 0.05). The decay of algal blooming color has been reported to occur during the daytime. For example, green snow s observed during periods of stable solar radiation (Hoham, 1975), and disappeared after the sun (Kawecka, 1986). Although snow algae require light for photosynthesis, a negative relationship has been reported between cell concentration and solar radiation. For example, light irradiation experiments have shown that an irradiation intensity of 95 μmol PAR m$^{-2}$ s$^{-1}$
(equivalent to 19 W m$^{-2}$) was most suitable for the sexual reproduction of snow algae (Hoham et al., 1998), and photoinhibition occurred at radiation intensities stronger than 200 μmol PAR m$^{-2}$ s$^{-1}$ (equivalent to 40 W m$^{-2}$) (Procházková et al., 2019a). Another study calculated that the snow depth (approximately 2 m), where snow algae were observed, was the layer



through which 0.1% of the solar radiation passed through wet snow (Curl et al., 1972). All snowpack layers used in this study were composed of wet snow. Assuming the same transmissivity of the snowpack as Curl et al. (1972), the intensity of solar

radiation is estimated to be 3–4% in the layers between II and III, where snow algae are distributed during the daytime. Based on these results, 30 W m$^{-2}$ or less solar radiation appears to be the optimal condition for snow algae activity. The negative phototaxis of motile algae also plays a role in determining their distribution. For example, algae (*Chlamydomonas nivalis*) changed movement velocity in response to light (Vladimirov et al., 2004). *Chlamydomonas* cells recognize light in their eyespots and exhibit either negative or positive phototaxis (Witman, 1993). In addition to radiation intensity, Latta et al. (2009)

pointed out the possibility of algae migrating to escape from grazers and predators. However, tardigrades and rotifers, which are potential predators of snow algae, started to move earlier than snow algae (between 5:00 and 8:00 for predators and between 8:00 and 11:00 for snow algae). It is unlikely that the presence of predators influenced the movement of snow algae.

The vertical distribution of chemical solutes indicated that nutrients were most abundant in the snow surface layer, which appears to be preferable for snow algae. The nutrients of snow algae, $NH_4^+$, $PO_4^{3-}$, and $K^+$, seemed to be supplied from litter

(Jones, 1991) and concentrated on the snow surface throughout the study period (Figs. 7, S5). This result implies that snow algae migrate upward to the surface layer at night, which is the preferred layer in terms of nutrients for algal growth. Therefore, the diel change in algal distribution is likely driven by the availability of nutrients and the intensity of solar radiation within the snowpack.

### 4.2.2 Factors determining vertical distribution of microinvertebrates

The timing of microinvertebrate migration coincided with changes in the intensity of solar radiation, suggesting that microinvertebrates also migrate vertically to avoid intense solar radiation in the snowpack. Downward movements of tardigrades and rotifers started at 5:00 when the solar radiation started to increase. They moved upward again at 14:00, when the solar radiation began to decline. The proportion of their population in layer II showed significant negative correlations with the intensity of solar radiation (tardigrades and solar radiation: r = -0.45, p = 0.02 < 0.05; tardigrades and temperature: r = -

0.10, p = 0.59 > 0.05, rotifers and solar radiation: r = -0.44, p = 0.02 < 0.05; rotifers and temperature: r = -0.28, p = 0.16 > 0.05). The photototaxis of microinvertebrates has been reported to vary depending on the species and growth stage (Beasley, 2001; Colangeli et al., 2019). Tardigrades of the genus *Hypsibius*, which is the same genus as the tardigrades found in this study, cannot survive in environments with a high intensity of ultraviolet light (Suma et al., 2020). Furthermore, the reproduction rate of rotifers decreased under the radiation intensity higher than 0.5 W m$^{-2}$ (Kim et al., 2014). Food movement

is another factor affecting migration. Although snow algae are likely to be their main food source, snow algae migrated to the lower layers later than invertebrates (Fig. S2), suggesting that feeding was not the reason for the movement of the tardigrades and rotifers. Furthermore, when the tardigrades moved downward during the day, some of their populations remained in the surface layer (layer I). This is probably due to the positive phototaxis of juveniles, for which photoreceptors have not yet been developed. One study suggested that the production of proteins (opsins) associated with photoreceptors differed among the

three life history stages of eggs, juveniles, and adults for tardigrades of the genus *Hypsibius* (Fleming et al., 2021). Furthermore,



a species of tardigrade, *Hypsibius convergens* exhibited positive phototaxis until the second molt (Baumann, 1961). Thus, the tardigrades in the snowpack appear to be sensitive to solar radiation and migrate with changes in intensity.

Results in this study suggest that the DVMs of snow-ice microbes occur in the following process: (i) during nighttime, snow algae concentrate on the nutrient-rich snow surface, and microinvertebrates concentrate where abundant algae; (ii) as solar
radiation increases, these microbes move downward to avoid intense solar radiation; (iii) when solar radiation begins to weaken, microbes move upwards again. Note that this study did not consider the effect of the physical structure within the snowpack, which also affects the distribution of microbes, as snow algae have been reported to concentrate in the snow just above the ice layers within a snowpack (Hoham, 1975).

## 4.3 Roles of DVM of snow-ice microbes in snowpack ecosystems

The distribution of microbes in the snow pits and core samples shows the microbial contrast of the snow layers above and below a depth of 30 cm; the snow layer above this depth is microbially active, whereas the layers below this depth are almost abiotic environments. More than 90% of each microbial species found in the snow core were concentrated in the surface layer above a depth of 30 cm. We propose that this surface layer refers to as the microbial active snow surface layer (MASS layer) in this study. The DVM of snow-ice microbes indicated the presence of distinctive material circulations driven by microbes
between nighttime and daytime within the MASS layer, such as photosynthesis, growth, feeding, and transport. At night, snow algae take up nutrients ($NO_3^-$, $NH_4^+$, and $PO_4^{3-}$) for their growth in the upper part of the MASS layer. When the intensity of solar radiation increases, they move to the lower part of the MASS layer and take up $CO_2$ for photosynthesis. Microinvertebrates feed on snow algae near the snow surface at night. Their movement to the lower part of the MASS layer during the day can transport materials downward. These materials, produced by microbial activity, first concentrate on the
surface with snowmelt, but some of the water-soluble materials would flow downward in the snowpack with meltwater.

The thickness of the MASS layers is likely to affect primary production and biogeochemistry in the snowpack, and may further affect the surrounding environment. Although the thickness was 30 cm in this study, it is likely to vary depending on the season, landscape, geographical location, and the microbial community. As microbial DVM is associated with the intensity of solar radiation, the MASS layer may be thicker during the season and in locations with more intense solar radiation. The
materials microbially produced in the MASS layers in snowpacks can be transported with meltwater to the soil, rivers, and lakes in the surrounding area (Antony et al., 2017; Domine, 2019). Therefore, quantification of the material cycles in the MASS layer is important not only for understanding snowpack ecosystems, but also for their impact on the surrounding environment (Falkowski et al., 1998; Field et al., 1998; Carpenter et al., 2005; Solomon et al., 2011). Further studies on microbial DVM and MASS layers in snowpacks are necessary to understand the biogeochemical roles of seasonal snow cover
in alpine and polar environments.



## 5 Conclusion

The DVMs of snow-ice microbes within a snowpack in an alpine forest in northern Japan were described based on periodic and cross-sectional sampling. Microbes observed in snowpacks can be divided into two groups: those with and without diel changes in their vertical distribution. The vertical distributions of type A and B snow algal cells, tardigrades, and rotifers

differed significantly between nighttime and daytime. In contrast, those of the Type C algal cells and *Chi. nivalis* did not change temporally, which concentrated at the surface snow layer throughout the study period. The difference in the temporal changes in the distribution can be explained by the motility of the microbes. The microbes without diel changes in vertical distribution were all immotile cells. Their distributions are likely to be determined mainly by the sources of the cells and the physical processes of snowmelt. The presence of the maximum population in the surface layer during the daytime suggests

that percolating meltwater does not affect their distribution within the snowpack. The microbes showing a diel change in vertical distribution were all motile and were distributed in the surface layer at night and in the lower layers during the daytime. These diel changes coincided with the intensity of solar radiation, suggesting that snow algae DVM occurred to avoid intense solar radiation. Analysis of chemical solutes in the snowpack revealed that $NH_4^+$, $PO_4^{3-}$, and $K^+$ were highest in the surface layer, indicating that the surface layer is favorable for snow algal growth in terms of nutrient availability. Snow algae probably

migrate to the surface layer to uptake nutrients during nighttime and to the lower layers to photosynthesize under proper radiation conditions during the daytime. The timing of changes in microinvertebrates (tardigrades and rotifers) also coincided with the intensity of solar radiation. They probably feed on snow algae at the surface layer at night and move to the lower layers to avoid intense solar radiation during the daytime.

The vertical distributions of microbes in the snow pits and core samples indicated that the abundance of microbes was
distinctive above and below a depth of 30 cm. The layers above this depth, MASS layer, are likely to store and circulate carbon and nitrogen produced by snow-ice microbes; thus, they play an important role in snowpack ecosystems. In this study, horizontal migration was not considered. To understand the roles of MASS layers in snowpack ecosystems, further studies on the three-dimensional migration of snow-ice microbes and the quantification of material cycles within snowpacks are necessary.

## Competing interests

The contact author has declared that none of the authors has any competing interests.

## Acknowledgement

Sampling and all analyses were supported by JSPS KAKENHI (19H01143, 20K21840, 20H00196, and 21H03612) and by the Arctic Challenge for Sustainability II (ArCS II, Program Grant Number JPMXD1420318865) for N. T. Making and publishing of the manuscript was supported by JSPS KAKENHI (22J11017, 22KJ0471, and 24KJ0118) for M. O.



## Author contributions

Sampling for research: M. O.; manuscript concept: M. O. and N. T.; microscopy: M. O.; chlorophyll a: M. O.; chemical analysis: M. O.; and statistical analysis: M.O. All authors have edited and reviewed the manuscript.

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
