# Peer review of "The diel vertical migration of microbes within snowpacks driven by solar radiation and nutrients"

_EGUsphere, 2024_

## Author Comment (AC1)

Dear Dr., Florent Domine,

Thank you very much for the review of our manuscript. We agree with the reviewer's point about the non-microbial aspects. We do not have data on the snow temperature or water content at the time of sampling, and no more sampling was conducted even in 2024. In this revision, we added a new Supplementary Figure S1 which took photos of every cross section of all snow samples to check for the distribution of ice layers. We hope that all of you agree. We also added detailed information based on the comments of the reviewer regarding the lack of detailed description of the method including the property of the snowpack, and the discussion. With these corrections, we believe that the revised version will be closer to the reviewer's ideal manuscript. Thank you once again for your consideration of our paper.

Sincerely,
On behalf of all authors

*Masato Ono*

---

## Author Comment (AC2)

**Reviewer 1:**

Thank you much for the review of our manuscript. We have addressed all the comments. Please see below for our point-by-point responses to the reviewers (in blue and preceded by "REPLY:").

The results presented in this study provide significant insights into snow algae blooms and their associated communities. The meticulous effort invested in sampling, separating, and analyzing various microorganisms under the microscope throughout a 24-hour cycle is commendable and underscores the study's value. It's important to acknowledge the inherent challenges of field studies, as opposed to laboratory experiments where there is much control of what is happening, which should be considered when evaluating this research. However, the study of these communities is conditionate by how these blooms occur in nature, making it impossible to sample the same community twice given their patchy distribution. This entails a series of limitations when planning the experiment and drawing up conclusions that should be considered.

-As the results are presented, the observed differences could be due to vertical movements in the community because of changes in temperature and radiation, but they could also be due to their heterogeneous distribution (on the surface and along the vertical profile). As an example of the patchy distribution, there are differences if we add total chl-a along the vertical profile for the two days at 2:00 AM (30% difference between days), which gives an idea that the same community is not being compared even if it has been sampled in almost the same place. It is not clear to me the methodology used for sample collection, whether the heterogeneity of these communities was considered, or if the variability observed at each time point reflects the variability within the snowpatch. A more thorough explanation of the methods would be beneficial.

REPLY: We collected the samples at different surface in each sampling as described in Line 112 for preventing affect the next sampling. At each sampling time, the three different surfaces were collected because it would be possible to perform statistical analysis. As can be seen from the results of monitoring the snow surface (Figure 8 in the original manuscript), even if sampling is done at the same location, it is unlikely that the same distribution will be reproduced after 24 hours. We added more detailed description of the methods.

-Differences, between days, in the vertical distribution of chl-a observed in the 02:00 AM samples (with maximum concentrations in the lower layer vs to the surface, respectively), suggest that extending the study over a longer period might lead to, perhaps, different conclusions. This indicates that additional sampling may be needed to model vertical movements within a snow

patch, and this could be an initial observation of such movements. This aspect deserves more thoughtful consideration in drawing the conclusions.

REPLY: We agreed that the vertical distribution may change over a longer period. The sampling conducted in this study could not reveal seasonal changes, then we mentioned this possibility in the discussion. We added a sentence "Also, additional sampling may be needed to model vertical movements within the snowpack including their seasonal change." in the discussion (before Line 391).

-Another point to consider, is that algae are classified into motile and non-motile categories, and conclusions are drawn based on this classification. However, this study does not provide evidence for their motility. While it is assumed that these organisms demonstrate active movement, no motility structures were identified through the extensive microscopic observation of the algae. It would be necessary for the authors to further explore this in the discussion.

REPLY: When counting algae on the slides, the flagellum was lost, and they were often observed missing the locomotor apparatus. However, we observed some motile individuals, then match cells with and without flagella referred previous research (Muramoto et al., 2010; Matsuzaki et al., 2015, 2020; Procházková et al., 2019) which has observed algae with the same morphology that possess flagellar structures. Additionally, we observed an individual of Type A that still had its flagellum.

  We added a sentence "Alternatively, observed some motile cells were matched with and without flagella referred previous research (Muramoto et al., 2010; Matsuzaki et al., 2015, 2020; Procházková et al., 2019) which has observed algae with the same morphology that possess flagellar structures." in Line 145.

  We changed a photo of this flagellated algal cell to Figure 4a as shown below, and changed its caption. Now you can read "Figure 4: Snow-ice microbes inhabit the snow. (a), (b) Snow algae Type A (LM), arrowheads indicate flagella, (c) Snow algae Type B (LM), (d) Snow algae Type C (LM, dormant state), (e) Tardigrade Hypsibius spp. (LM), (e1) Skin of Hypsibius nivalis (PCM), (e2) Skin of Hypsibius sp. (PCM), (f) Rotifer Philodinidae gen. sp. (LM), (g) Fungi Chionaster nivalis (PCM). All scale bars are in micrometers.".

[Figure]

Revised Figure 4

We also added the sentences "Flagella were observed in some individuals as shown in Figure 4a." in Line 190, and "which were considered to flagellate algae including various species" in Line 287. Now you can read "There were two different sizes in this type of algae; small ones ranged from 4.6 to 9.9 μm (mean ± SD: 8.7 ± 1.5 μm, n = 245), and large ones ranged from 10.0 to 22.6 μm (14.8 ± 2.4 μm, n = 372) in length, respectively (Fig. 4a, b). Flagella were observed in some individuals as shown in Figure 4a. Type B had oval-shaped cells with green chloroplasts (Fig. 4c).", and "This group includes Types A and B snow algae which were considered to flagellate algae including various species (Muramoto et al., 2010; Matsuzaki et al., 2015, 2019, 2020; Procházková et al., 2019), tardigrades, and rotifers.", respectively.

-An important aspect for this study is understanding the properties of the snow, which have not been analyzed throughout the vertical profile. The snowpack conditions are crucial for explaining potential movements of microorganisms or defining a microbial active snow surface (MASS) layer. If data such as temperature and water content along this vertical profile have not been measured, the authors should provide evidence demonstrating that all samples exhibit uniform conditions and, for example, there is no ice layer restricting some of these vertical movements.

REPLY: Unfortunately, we do not have data on snow temperature or water content at the time of sample collection in this study. However, we took photos of every cross section at each sampling time. We observed the ice layers in some sampling time. As for the property of the snow, we confirmed that it is granular snow except for the ice layers, and in an observation of the cross section taken at 10 a.m. on May 4th prior to collecting the samples in this study, the snow temperature was consistently 0.1°C for the five layers

taken in this study. Recorded air temperature was continuously above freezing point, then it was impossible that the snow temperature drops below freezing. Based on these facts, we believe there is no gradient of snow temperature even during the sampling period.

We added the sentences "In the observation conducted prior to this study at 10:00 a.m. on May 4th, the snow temperature and the property of snow in five layers were 0.1°C and granular snow, respectively." before Line 112, and "Also, additional sampling may be needed to model vertical movements within the snowpack including their seasonal change." in the discussion (before Line 391).

Ln1. Title: The presented data is insufficient to substantiate this claim. Nutrient levels at 15cm might provide adequate conditions for algae to thrive, potentially not influencing their vertical movements. Please consider modifying the title accordingly.

REPLY: We still thought nutrients is an important factor for microbial vertical migration. The vertical distribution of chemical solutes every three hours as shown in Supplementary Figure S6, $PO_4^{3-}$ and $K^+$ at nighttime were near zero (0-1 µEq $Kg^{-1}$) in 3-13 cm in depth while that of more than 10 µEq $Kg^{-1}$ in 0-3 cm in depth. Therefore, we concluded that the snow surface is the more suitable for algal growth, then moved up to snow surface during nighttime. As another reviewer had pointed out, we added "in northern Japan" to emphasize that the results of this study are applicable only to Japan. Now you can read "The diel vertical migration of microbes within snowpacks in northern Japan driven by solar radiation and nutrients".

Ln 37-38. The use of degrees Kelvin in a study like this seemed unusual to me. To make it more universal, I would recommend changing the units to Celsius.

REPLY: We corrected as reviewer suggested Kelvin to Celsius written between Line 36-39. Now you can read "The climate model projected that the timing of snowmelt in the mountainous areas of central Japan would begin half a month earlier than the present climate when the global air temperature is 2℃ warmer than that in the pre-industrial period and that the snowpack would disappear two months earlier when it is 4℃ warmer (Kawase et al., 2020).".

Ln 98-102. Do the authors have information on the slope, water content, and temperature in the snow?

REPLY: As we mentioned in our reply to the previous comment, there is no data on water content, snow temperature, and slope. We revised the methods and discussion as we described previous answer to your general comment "-An important aspect for this study

is understanding the properties of the snow, which have not been analyzed throughout the vertical profile.-".

Ln 105. How did the authors decide where to take the samples?
I expect there's significant variability in the presence of algae across both the surface and vertical profile of the snow patch. This suggests that differences in the vertical distribution of the community were already present at the beginning of the experiment, under identical radiation and temperature conditions.

REPLY: Many green snow patches were observed at the study site. Samples were taken randomly and always different surface at each sampling time. As reviewer pointed out, there was variation in concentration of microbes between snow patches. Since we focused on the vertical distribution of their relative abundance within the snowpack, we expressed the results as a percentage not as a concentration. We added the sentences "randomly selected" in Line 111, and "In order to eliminate the bias in the average value due to the heterogeneity of the snow patches," in Line 152-153. Now you can read "Sampling was performed on three different surfaces randomly selected at each time point.", and "In order to eliminate the bias in the average value due to the heterogeneity of the snow patches, their vertical distributions are represented as proportions of cell concentrations in each layer of the snow pit or snow core.", respectively.

Ln 115. There appears to be an ice layer below 18 cm that could potentially disrupt the vertical movement of microbial communities. I'm concerned the presence and variability in ice layer depth across the snowpatch could impact some of the conclusions of this study. Including photos of each sample would greatly clarify whether the ice is affecting the vertical flow.

REPLY: We agreed reviewer's point that the presence of the ice layers certainly affect the vertical distribution of microbes. However, this point was not taken into consideration in this study. We added this possibility to the discussion "Furthermore, the property of snow was granular snow in all layers, then it was thought that the water content had no effect on the vertical distribution of the microbes. However, multiple ice layers were observed in the cross section of the snowpack (Fig. 1, S1). The presence of such ice layers may affect the vertical distribution of microbes, whereas the presence of ice layers was not considered in this study. Therefore, further investigation is required." in Line 337.

We also took photos of every cross section at each sampling time, and added these photos as a new Supplementary Figure S1 shown below. We observed the ice layers in some sampling time. For describe this information, we added sentences "Ice layers were observed in some sampling times, otherwise the grain shape of snow was granular snow

[Figure]

New Supplementary Figure S1: Cross-section of snowpack at different three locations for each sampling time.

Ln 130. Why did the authors choose not to filter/process the samples before freezing them? Freezing samples before processing can alter the nutrients present, potentially lysing millions of bacteria and resulting in a misunderstanding of the actual conditions in the snow. This consideration should be acknowledged in the discussion.

REPLY: We agreed, and added the points reviewer has raised to the discussion "One should note that freeze-thaw events, such as those involved in sample processing in this study, can alter chemical conditions primarily due to the activity of heterotrophic bacteria. Future studies should take this into consideration." in Line 343.

Ln 136. From the photos, it seems there is accumulated tree biomass on the snow patch surface. Was this biomass removed before aliquoting the samples? If not, it might have heightened surface chlorophyll levels. If it was removed, how was this process carried out? This could be a significant source of microfauna.

REPLY: The samples were sieved to remove any large impurities before sample analysis. This has not been explained in the methods, then we added "filtered through a sieve to remove the plant litter, then" in a sentence "After melting, the samples were separated into 10 mL for snow algae and fungi, 5 mL for soluble ion analysis, and the remainder for

microinvertebrates." in Line 130-132. Now you can read "After melting, the samples were filtered through a sieve to remove the plant litter, then separated into 10 mL for snow algae and fungi, 5 mL for soluble ion analysis, and the remainder for microinvertebrates.".

Ln 187. Could these be different species? Even if they are the same species, would it be worthwhile to analyse them separetly within the vertical profile? Variations in size like this might indicate differing abilities for vertical movement.

REPLY: Several species of algae with similar morphology have already been reported in this region (e.g., Muramoto et al., 2010; Matsuzaki et al., 2019), making it unlikely that they are the same species. Therefore, we believe that it would be worthwhile to analyze them separately within the vertical profile. This explanation is supported by our previous response, where we added the sentence "which were considered to include flagellate algae of various species" in Line 287. Now you can read "This group includes Types A and B snow algae which were considered to flagellate algae including various species (Muramoto et al., 2010; Matsuzaki et al., 2015, 2019, 2020; Procházková et al., 2019), tardigrades, and rotifers.".

Ln 195. It would be interesting to include images of both species of tardigrades to offer a clearer description of the community.

REPLY: The differences between these species are defined by the structure of their skins. Therefore, we changed the description of tardigrades in Line 194-195 and caption of Figure 4 with adding photos that showed the differences in their skins. Now you can read "Two species of tardigrades, *Hypsibius nivalis* and *Hypsibius* sp. (Ono et al., 2022), featured their skin (*Hypsibius nivalis*: reticular, *Hypsibius* sp.: smooth) dominated (Fig. 4e)." and "Figure 4: Snow-ice microbes inhabit the snow. (a), (b) Snow algae Type A (LM), arrowheads indicate flagella, (c) Snow algae Type B (LM), (d) Snow algae Type C (LM, dormant state), (e) Tardigrade *Hypsibius* spp. (LM), (e1) Skin of *Hypsibius nivalis* (PCM), (e2) Skin of *Hypsibius* sp. (PCM), (f) Rotifer Philodinidae gen. sp. (LM), (g) Fungi Chionaster nivalis (PCM). All scale bars are in micrometers.", respectively.

Ln 232. It seems that this section its already included in the title of the previous section. Please delete this section or modify the title of the preceding section.

REPLY: We deleted as reviewer suggested.

Ln 246. Could be associated to the different species?

REPLY: We did not count them while distinguishing between them for each sample.

However, since the ratio of these species was approximately 95:5 for *Hypsibius* sp.: *Hypsibius nivalis*, *Hypsibius* sp. was dominated the samples, and it is unlikely that there was a difference in distribution due to species. We added this information in the results "Mostly *Hypsibius* sp. was dominated the samples (*Hypsibius* sp. : *Hypsibius nivalis* = 95 : 1)." in Line195. Now you can read "Two species of tardigrades, *Hypsibius nivalis* and *Hypsibius* sp. (Ono et al., 2022), featured their skin (*Hypsibius nivalis*: reticular, *Hypsibius* sp.: smooth) dominated (Fig. 4e). Mostly *Hypsibius* sp. was dominated the samples (*Hypsibius* sp. : *Hypsibius nivalis* = 95 : 1). Identifying the rotifer specimens was difficult because of the absence of live species; however, Philodinidae dominated (Fig. 4f).".

Ln 276. This is very cool and provides valuable information for better understanding the dynamics of these communities. It appears that the green snow patch never reached the color intensity seen the first time...

REPLY: Not only the color in the photo, but also the upward movement of the snow algae changes depending on weather conditions, therefore, it is unlikely that the color completely return. Additionally, the purpose of taking this time-lapse photo was to confirm that there is no horizontal movement of microbes. For emphasizing it, we added a sentence "without spreading horizontally" in Line 276-277. Now you can read "The green color gradually disappeared without spreading horizontally by 9:00 in morning and remained until 17:20.".

Ln 290. Did the authors detect any motility structures in this group of cells that would better support this assumption?

REPLY: This overlaps with the previous answer. Please check the answer to your general comment "-Another point to consider, is that algae are classified into motile and non-motile categories, and conclusions are drawn based on this classification-".

Ln 293. Dormant cell or a cell in a state prior to dormancy?

REPLY: Since it was a dormant cell, we would like to avoid this confusion by deleting "algae" from "a dormant algal cell". Now you can read "The microbes without diel changes included Type C algae and *Chi. nivalis*, which are a dormant cell and a fungus, respectively, and both of them are immotile.".

Ln 305. Were the cells found in aggregates or individually? This could explain why they are not washed from the surface.

REPLY: When observed under a microscope, these algae were individual and not in aggregates. Therefore, it seems unlikely that aggregates could have prevented the washout from the snow surface.

Ln 312. Could it be that meltwater washes the algae cells and that they actively return to the surface when melt rates decrease due to changes in radiation? Type C may be producing exopolysaccharides that allow the formation of larger groups of cells that limit their vertical movement

REPLY: As describes at the beginning of the discussion in Line 291-305, not only Type C but also immobile fungi did not move downward, then we believe that they were not washed by meltwater.

Ln 329. The water content in the snow and the presence of ice layers are likely crucial factors for explaining the vertical movement of microorganisms.

REPLY: As we mentioned in our previous response, we acknowledged that the ice layers could affect the distribution of microbes. We have added sentences addressing this topic, consistent with our reply to your comment, "Ln 115. There appears to be an ice layer below 18 cm that could potentially disrupt the vertical movement of microbial communities.-".

Ln 358. Did the authors see differences in the distribution of the two species of tardigrades along the vertical profile?

REPLY: This overlaps with the previous answer. Please check the answer to your specific comment "Ln 246. Could be associated to the different species?".

Ln 365. As previously mentioned, could it be that the meltwater is washing them downward?

REPLY: As we mentioned in previous answer, not only Type C but also immobile fungi did not move downward, then we believe that they were not washed by meltwater.

Ln 369. My concern is that an ice layer might be causing this difference. Since all the samples were collected from the same snow patch, which has a similar slope, this ice layer could be present throughout the entire area. Thus, this could imply that without an ice layer, there are no barriers to vertical movement, making it unnecessary to discuss a microbial active snow surface layer.

REPLY: As we mentioned in our previous response, we acknowledged that the ice layers could affect the distribution of microbes. However, the newly submitted Supplementary Figure S1 suggests that the ice layers do not seem to act as a barrier to the vertical

distribution of microbes. Therefore, we believe that the concept of DVM presented in this study cannot be dismissed. We have added sentences addressing this topic, consistent with our reply to your comment, "Ln 115. There appears to be an ice layer below 18 cm that could potentially disrupt the vertical movement of microbial communities.-".

Ln 391. Please consider revising the conclusions based on the earlier suggestions.

REPLY: Based on the reviewer's comments, the conclusion of this study remains unchanged, however, the words added stating that the physical properties of snow need to be taken into consideration in further study. We added the words "the consideration of physical properties such as the presence of ice layers," in Line 447-449. Now you can read "To understand the roles of MASS layers in snowpack ecosystems, further studies on the consideration of physical properties such as the presence of ice layers, the three-dimensional migration of snow-ice microbes, and the quantification of material cycles within snowpacks are necessary.". As another reviewer pointed out, we added the words "in northern Japan" in Line 411 to emphasize the fact that our findings can be applied to Japan. Now you can read "The layers above this depth, MASS layer, are likely to store and circulate carbon and nitrogen produced by snow-ice microbes; thus, they play an important role in snowpack ecosystems in northern Japan.".

Figure 7. It may be a good idea presenting the vertical profiles of NH4, PO4, and K in the first row, as they share a similar profile distinct from the others.

REPLY: As reviewer pointed out, we put those three components at the top of the figure.

[Figure]

Revised Figure 7

---

## Author Comment (AC3)

**Reviewer 2:**

Thank you much for the review of our manuscript. We have addressed all the comments. Please see below for our point-by-point responses to the reviewers (in blue and preceded by "REPLY:").

The manuscript by Ono et al., presents interesting data on the vertical distribution and dispersal of algae and microscopic invertebrates in the snow patches. Snow ecosystems are poorly known in general, therefore each piece of data on the algae and their consumers are crucial. Although interesting, manuscript cannot be accepted in the present form. Manuscript require corrections and changes of the tone in the interpretation of the results. I like the idea of MASS and in my opinion the idea makes sense, at least for algae and invertebrates. However, without some clarifications (e.g. sampling) the evaluation of the data robustness is difficult.

REPLY: As the reviewer pointed out, we revised the method to include more details especially in sampling.

- I would mute the tone of the results and discussion. The studies were conducted in the Japanese forests, on the mountain slope in the specific insular climate, it is hard to extrapolate these data to other snow ecosystems. It will be better to highlight in many places that results are valid for Japan. Therefore, I support the importance of the findings but some sentences are overstated.

REPLY: In the title, discussion, and conclusion, we would like to state that the findings of this study are only applicable to Japan and need further studies to confirm the presence of MASS layers in other countries. We added the words "in northern Japan" in Title and Line 411. Now you can read "The diel vertical migration of microbes within snowpacks in northern Japan driven by solar radiation and nutrients", and "The layers above this depth, MASS layer, are likely to store and circulate carbon and nitrogen produced by snow-ice microbes; thus, they play an important role in snowpack ecosystems in northern Japan.", respectively. We also revised the sentences "Further studies on microbial DVM and MASS layers in snowpacks are necessary to understand the biogeochemical roles of seasonal snow cover in alpine and polar environments." in Line 388-390 to "Further studies on microbial DVM and MASS layers in snowpacks are necessary. These studies should first confirm the presence of the MASS layer in snow not only in northern Japan but also in other countries, and then aim to understand the biogeochemical roles of seasonal snow cover in alpine and polar environments.".

- Could you compare radiation in the forests and open landscape? I'm not convinced that UV could be explanation of the migration in the forests (even without leafes trees offer sort of protection).

REPLY: We do not have the data of UV while measured solar radiation outside the forest. We added the data of solar radiation outside the forest in Figure 3b, and revised its caption "Figure 3: Meteorological conditions recorded during the study period. (a) air temperature, (b) solar radiation. Solar radiation recorded inside the forest is shown as a black line while that recorded outside the forest is shown as a gray line. The period of nighttime is shown as gray zones.".

[Figure]

Revised Figure 3

The data showed that the intensities of solar radiation were lower inside the forest. We added the description and data regarding solar radiation recorded outside the forest in Line 100-102, 177-180, and 315. Now you can read "The intensity of solar radiation inside and outside the forest was recorded every 10s using a pyranometer (ML-020VM, EKO, Japan) with a data logger (LR5091, HIOKI, Japan) at a height of 5 cm from the snow surface by setting it on a small pedestal.", "Solar radiation ranged from 0 to 755 W

m$^{-2}$ inside the forest, and 1 to 938 W m$^{-2}$ outside the forest (Fig. 3b). The intensity of solar radiation started to increase at 6:00, reached its peak at 14:00 inside the forest (755 W m$^{-2}$), 12:00 outside the forest (938 W m$^{-2}$), respectively, and then decreased continuously until 20:00, while that of outside the forest. In this study, daytime was defined as the period from 9:00 to 16:00 when the intensity exceeded 150 W m$^{-2}$ inside the forest, and nighttime was defined as the remaining period of the day.", and "After the maximum intensity of solar radiation at 14:00 (755 W m$^{-2}$) inside the forest, the position of the maximum population changed again to the upper layer (layer I), suggesting that the algae moved upward over time.", respectively.

We concluded that even these intensities were too strong for microbes. The results we cited in the discussion (Line 324-327) were much lower than the intensities observed in this study. We also added a sentence "The daytime intensity of solar radiation recorded in this study, both inside and outside the forest was higher than that shown in previous studies." in Line 327. Now you can read "For example, light irradiation experiments have shown that an irradiation intensity of 95 μmol PAR m$^{-2}$ s$^{-1}$ (equivalent to 19 W m$^{-2}$) was most suitable for the sexual reproduction of snow algae (Hoham et al., 1998), and photoinhibition occurred at radiation intensities stronger than 200 μmol PAR m$^{-2}$ s$^{-1}$ (equivalent to 40 W m$^{-2}$) (Procházková et al., 2019a). The daytime intensity of solar radiation recorded in this study, both inside and outside the forest was higher than that shown in previous studies. Another study calculated that the snow depth (approximately 2 m), where snow algae were observed, was the layer through which 0.1% of the solar radiation passed through wet snow (Curl et al., 1972).".

- I would be happy to see discussion on another option/reason of migration between snow layers. I would expect that predators (e.g. springtails) could impact on the migration; preys avoid predators (birds and ice worms on the North American glaciers are a good example of such relation).

REPLY: We agreed that the presence of predators could be affect the microbial vertical distribution. We addressed this in the discussion by citing papers which described several microinvertebrates in the snowpack (Hanzelová et al., 2018; Yakimovich et al., 2020) and added the sentences "In this study, other predators, such as springtails which have observed in previous study (Hanzelová et al., 2018; Yakimovich et al., 2020), were not counted. Therefore, there remains a possibility that the observed behavior is a response to predation pressure. Future research should also consider the impact of organisms that prey on microinvertebrates." in Line 368.

- it is not clear, how many cores were collected at each spot? One core = one sample. What about subsamples = layers? what about replicates = sampling of the same layers?

REPLY: As describes in the method in Line 120-122, one core was collected (at 5:00 on the May 7th). Three different surfaces (locations) were collected at each sampling time as described in Line 112. In each surface, we collected the samples from an area of 5 × 5 cm in five layers across snow depths as described in Line 110-112.

To prevent confusing, we deleted a word "subsample" in Line 132, 141, and 147, and modified a sentence about the snow core sampling in Line 120-122. Now you can read "The samples for snow algae and fungi were stored in 10 mL plastic tubes with 3% formaldehyde.", "Twenty microliters of the sample were transferred from the sample onto a glass slide.", "5–200 µL from the samples were injected into a filter holder equipped with a 0.45 µm PTFE membrane filter (JHWP01300, Merck Millipore, Germany), then filtered using a pump (Linicon LV-125, Nitto Kohki, Japan).", and "The snow core sample was collected at 5:00 a.m., immediately after sunrise on May 7th. The core, with a total length of 113 cm, was cut horizontally every 10 cm using a snowsaw and preserved in Whirl–Pak bags.", respectively.

- How many cores? How many pits?

REPLY: Snow pits were collected on three surfaces for each time period, and a snow core collected only once (5:00 on May 7th).

- Why algae were not identified to genus or family level? Even though many snow algae are morphological species complexes, still their identification at higher taxonomic level is possible.

REPLY: We agreed reviewer's point. We identified the genus level (*Chloromonas*), then described at the genus level in Line 186. Now you can read "Three morphological types of *Chloromonas* snow algae were dominant in the snowpack (snow algae types A–C) (Fig. 4a–d).". We have already described about classification of algae in Line 144-145, then added the words "species level". Now you can read "Algal species were not identified species level in this study, because morphological taxa are not always represented as phylogenetic species of snow algae".

- Why authors did not use the best predictor (e.g. by PCA approach), then focus on the analysis involving the most important variables.

REPLY: We have conducted PCA and CCA approaches but the results were unclear, and we believe our current description is clearer.

- Im not sure that tardigrades are migrating between layers. It seems they are splitting from one big group into upper and lower groups. It could be accidental?

REPLY: The reviewer is correct in pointing out that Figure 5 shows a separation between the upper and lower groups. However, looking at the vertical distribution of tardigrades at 8:00 in Supplementary Figure S2, only a few individuals are concentrated on the snow surface. This suggests that tardigrades also moved vertically. The high proportion of observations on the surface at 11:00 and 14:00 is based on the fact that a few individuals were found only on the snow surface, which is reflected in the percentages.

- Finally, I miss basic data on the general dispersal features of algae and micorinvertebrates. Do we know what is the avarege distance the target organisms can disperse withing specific time? At least any basic data could help in the convincing reviewers to observed dispersal.

REPLY: For examples, the average swimming speed of *Chloromonas* algae (*Chloromonas reinhardtii*) was 106 μm s$^{-1}$ (Liu et al., 2020) and the average walking speed of *Hypsibius* tardigrade (*Hypsibius exemplaris*) was 163 μm s$^{-1}$ (Nirody et al., 2021). We added this information in discussion "These microbes were presumed to move tens of centimeters in the snow within a few hours. For example, *Chloromonas reinhardtii*, a species similar to snow algae, has been shown to move at an average speed of 106 μm s$^{-1}$ (Liu et al., 2020), and *Hypsibius exemplaris*, a genus related to tardigrades found in the snow, has been shown to move at an average speed of 163 μm s$^{-1}$ (Nirody et al., 2021)." between Line 287 and 288.

Abstract:
I suggest to higlight that percolation do not impact the results.

REPLY: We added a sentence "without washed out by percolated meltwater" in Line 13. Now you can read "Other microbes, including algal spores and fungi, remained on the surface layer throughout the day without washed out by percolated meltwater.".

Introduction:

Line 26: i.a. tardigrades and rotifers

REPLY: We corrected as reviewer suggested in the revised manuscript. Now you can read "These animals include i.a. tardigrades, rotifers (Hanzelová et al., 2018; Yakimovich et al., 2020; Ono et al., 2021, 2022), springtails (Hao et al., 2020), and winter stoneflies (Negoro, 2009).".

Line 27: add reference

REPLY: We added a reference (Ono et al., 2021), then now you can read "They feed on algae and redistribute organic matter as they migrate through the snowpack (Ono et al., 2021).".

Line 30: tropically? typo

REPLY: We meant "trophically". We corrected it, then now you can read "Because these organisms are trophically associated with each other (Brown et al., 2015; Ono et al., 2021), snowpacks can be acknowledged as unique ecosystems (Domine, 2019).".

Line 37: use one unit (C or K)

REPLY: We corrected as reviewer suggested by using °C. Now you can read "The climate model projected that the timing of snowmelt in the mountainous areas of central Japan would begin half a month earlier than the present climate when the global air temperature is 2°C warmer than that in the pre-industrial period and that the snowpack would disappear two months earlier when it is 4°C warmer (Kawase et al., 2020).".

Lines 45: can author say mating in term considered to flagellate algae including various species s of algae?

REPLY: There is nothing specific that can be described about mating in this study. However, the term "algal growth" used in the discussion (in Line 341 and 404) including their mating. We added the words ", including their mating" in Line 341. Now you can read "This result implies that snow algae migrate upward to the surface layer at night, which is the preferred layer in terms of nutrients for algal growth, including their mating.".

Line 64: reference is missing

REPLY: We added references (Engstrom et al., 2020; Ono et al., 2021), then you can read "Tardigrades and rotifers have often been observed feeding on snow algal cells in snowpacks (Ono et al., 2021) and may affect the distribution of snow algae, although there have been few reports on the association between snow algae and consumers (Engstrom et al., 2020; Ono et al., 2021).".

Results:
Line 193: not genera. Phyla.

REPLY: We corrected as reviewer suggested. Now you can read "Two phyla (Tardigrada and Rotifera) of microinvertebrates and snow fungi were also

frequently observed in the snowpack (Fig. 4e–g).".

Line 195: specimens instead of species

REPLY: We corrected as reviewer suggested. Now you can read "Identifying the rotifer specimens was difficult because of the absence of live species; however, Philodinidae dominated (Fig. 4f).".

Figure 4: add names of higher taxonomic levels before names of species, for example Tardigrade Hypsibius spp.

REPLY: We added higher taxonomic levels for tardigrade, rotifer, and fungi. Now you can read "Figure 4: Snow-ice microbes inhabit the snow. (a), (b) Snow algae Type A (LM), arrowheads indicate flagella, (c) Snow algae Type B (LM), (d) Snow algae Type C (LM, dormant state), (e) Tardigrade *Hypsibius* spp. (LM), (e1) Skin of *Hypsibius nivalis* (PCM), (e2) Skin of *Hypsibius* sp. (PCM), (f) Rotifer Philodinidae gen. sp. (LM), (g) Fungi Chionaster nivalis (PCM). All scale bars are in micrometers.".

Line 219: 26:00??

REPLY: In the methods, a sentence "In this study, the period from 0:00 on May 6th to 0:00 on May 8th was represented as 0:00 to 48:00." was described in Line 113-114 in original manuscript. If this description is misleading, we can correct in the revised manuscript.

Figure 5: For tardigrades, rotifers and fungi the names of axis are missing.

REPLY: We added "%"at the axis as reviewer suggested. The revised Figure 5 was shown below.

[Figure]

Revised Figure 5